# A re-appraisal of the ENSO response to volcanism with paleoclimate data assimilation

Feng Zhu 1,2, Julien Emile-Geay 2✉, Kevin J. Anchukaitis 3,4,5, Gregory J. Hakim 6, Andrew T. Wittenberg 7, Mariano S. Morales 8,9, Matthew Toohey 10 & Jonathan King 3,5

The potential for explosive volcanism to affect the El Niño-Southern Oscillation (ENSO) has been debated since the 1980s. Several observational studies, based largely on tree-ring proxies, have since found support for a positive ENSO phase in the year following large eruptions. In contrast, recent coral data from the heart of the tropical Pacific suggest no uniform ENSO response to explosive volcanism over the last millennium. Here we leverage paleoclimate data assimilation to integrate both tree-ring and coral proxies into a reconstruction of ENSO state, and re-appraise this relationship. We find only a weak statistical association between volcanism and ENSO, and identify the selection of volcanic events as a key variable to the conclusion. We discuss the difficulties of conclusively establishing a volcanic influence on ENSO by empirical means, given the myriad factors affecting the response, including the spatiotemporal details of the forcing and ENSO phase preconditioning.

[1] School of Atmospheric Sciences, Nanjing University of Information Science and Technology, Nanjing, China. [2] Department of Earth Sciences, University of Southern California, Los Angeles, CA, USA. [3] Laboratory of Tree-Ring Research, University of Arizona, Tucson, AZ, USA. [4] School of Geography, Development, and Environment, University of Arizona, Tucson, AZ, USA. [5] Department of Geosciences, University of Arizona, Tucson, AZ, USA. [6] Department of Atmospheric Sciences, University of Washington, Seattle, WA, USA. [7] NOAA Geophysical Fluid Dynamics Laboratory, Princeton, NJ, USA. [8] Instituto Argentino de Nivología, Glaciología y Cs. Ambientales, Consejo Nacional de Investigaciones Científicas y Técnicas (CONICET), Mendoza, Argentina. [9] Laboratorio de Dendrocronología, Universidad Continental, Huancayo, Peru. [10] Institute of Space and Atmospheric Studies, University of Saskatchewan, Saskatoon, SK, Canada. ✉email: julieneg@usc.edu

The El Niño-Southern Oscillation (ENSO), the quasi-periodic alternation of warm and cold phases of the tropical Pacific ocean-atmosphere system, is the leading source of global interannual climate variability[1]. ENSO influences weather conditions not only in the tropical Pacific[2], but also globally through atmospheric teleconnections[3]. Skillful prediction of the ENSO cycle, including its phase and amplitude, is therefore key to the successful forecasting of worldwide meteorological and oceanographic conditions at sub-seasonal to seasonal scales.

External forcing has the potential to affect ENSO variability[4–7]. In particular, explosive volcanism may inject large amounts of sulfate aerosols into the atmosphere, abruptly reducing incoming shortwave radiation and affecting the subsequent global ocean-atmosphere climate variability for several years[8]. A causal relationship between large eruptions and ENSO would be important for evaluating climate models' sensitivity to volcanic forcing, as well as assessing the risk of geoengineering solar radiation management schemes that emulate the stratospheric sulfate aerosol loading characteristic of large explosive eruptions[9]. The link between volcanism and ENSO has been vigorously debated since it was first proposed[10]. Since then, several tree-ring based observational studies have found support for an El Niño-like response in the year following large eruptions[4,11–14] and at least five mechanisms have been proposed to account for this relationship: (i) the ocean dynamical thermostat (ODT)[5,15,16], which states that the upwelled water in the eastern Pacific (EP) makes the region less sensitive to radiative forcing than the western Pacific, and leads to nonuniform Pacific SST response to uniform incoming solar radiation reductions after eruptions; (ii) the land-ocean temperature gradient (LOTG) mechanism[6,17,18], which states that the low thermal inertia of the land introduces a LOTG after eruptions, which affects the Pacific zonal wind anomalies and hence the ocean temperature; (iii) the subtropical wind stress curl mechanism[19,20], which states that the initial enhanced cooling in EP after eruptions leads to a negative (anticyclonic) subtropical wind stress curl that drives equatorward convergence of warmer subtropical waters and delays the ODT; (iv) the tropical teleconnection mechanism[18,21], which states that volcanically induced cooling of tropical Africa weakens the West African monsoon and alters the Walker circulation; and (v) the Inter Tropical Convergence Zone (ITCZ) shift mechanism[20–23], which states that the Northern Hemisphere (NH) cooling after NH eruptions shift the ITCZ southward, which weakens the Pacific trade winds and leads to an El Niño-like response. Several studies have also suggested a La Niña-like response 2 years after an eruption[14,24], which could be due to the oscillatory nature of ENSO dynamics[14], or to the eastward position of the anomalous western North Pacific anticyclone, exciting upwelling Kelvin waves and enhancing the thermocline and zonal advection feedbacks, leading to a greater cooling rate in the eastern Pacific[24].

Ensemble simulations with a highly simplified ENSO model[16] suggested a threshold effect that would make ENSO insensitive to all but the largest eruptions of the past millennium (approximately the magnitude of Krakatau and above). This has generally been confirmed by experiments with more complex models[20–23,25]. However, a recent analysis of a long, monthly coral record from the heart of the tropical Pacific[26,27] suggests no uniform ENSO response to all eruptions over the last millennium, even for the largest eruptions[28]. This is consistent with results from recent modeling studies using large ensembles that allow quantification of the influence of stochastic, as well as deterministic, elements[25]. Indeed, ENSO is thought to be affected by multiple uncertain or poorly constrained factors, including the phase of the quasi-biennial oscillation (QBO)[29], the forcing

magnitude, location, and season of the eruption[20,25], as well as pre-conditioning of the ENSO state (neutral, Central Pacific El Niño, Eastern Pacific El Niño, or La Niña)[17]. Yet, observational studies are hindered by the limited number of well characterized eruption events, the temporal resolution of volcanic forcing reconstructions, and the spatial and temporal availability of proxy records.

Paleoclimate records offer a longer period of observation but conflicting accounts: reconstructions based mostly on tree-ring proxies[4,11–14], which experience ENSO through remote teleconnections, have been used to argue of an El Niño-like response within a year of the eruption; in contrast, reconstructions using corals from the core ENSO region[27,28,30] – which provide a local, albeit discontinuous, record of ENSO variations – do not support this conclusion.

In this study, we re-appraise the potential links between volcanism and ENSO, by integrating the latest paleoclimate evidence from both tree rings and corals into the consistent dynamical framework of paleoclimate data assimilation (paleo-DA)[31,32], and interpret the results in the context of recent modeling work showing a large role for both initial and boundary conditions in shaping the climate response to volcanism[17,25].

## Results

**Corals vs tree rings.** Coral archives are a natural choice for ENSO reconstruction due to their geographical proximity to and within ENSO centers of action[33] and the demonstrated link between the geochemistry of their skeletons and ENSO conditions[34–36]. However, their limited time span produces discontinuous records, which can only be pieced together by splicing individual coral records[27]. The longest and most complete coral ENSO record published to date[27,28,37] is located at Palmyra atoll, at the edge of the Niño 3.4 region. The record covers 535 of the past 1000 years, leaving many gaps, though with good replication over intervals of overlap. Tree-ring based proxies, on the other hand, have been used to build long, precisely dated, and heavily replicated reconstructions that continuously span the Common Era. Their distance to ENSO centers of action means that they rely on teleconnections between the tropical Pacific and their local (terrestrial) rainfall and temperature anomalies, leaving them vulnerable to confounding factors. Combining data from both archives could ameliorate their individual limitations. Before doing so, however, we compare reconstructions assimilating the two archives separately, as this provide insights into the possible causes of the discrepancy between previous studies and on how to interpret a combined signal.

Our strategy leverages the paleoclimate data assimilation (paleo-DA) algorithm of the Last Millennium Reanalysis (Methods). We first assimilate corals to reconstruct tropical Pacific surface temperature during the boreal winter (DJF; we label the time axis of the reconstructions based on the calendar year of January, following existing conventions[11,12,33,38]). Our coral collection includes the synthesis of Ocean2k[30,39], supplemented by the latest Palmyra record[28]. To mitigate data attrition, only records that reach back before 1750 CE are assimilated. The resultant reconstruction is denoted as LMR (Corals), and its spatiotemporal skill is presented in Fig. 1a, d. The skill of the Niño 3.4 reconstruction is high compared to many other existing reconstructions (Supplementary Fig. 1), with a coefficient of determination ($R^2$) of 0.47 and coefficient of efficiency (CE) of 0.45 against ERSSTv5 (Extended Reconstructed Sea Surface Temperature v5)[40]. However, due to the temporal gaps in the Palmyra record, the reconstruction is incomplete over the last millennium (Fig. 2b) and the ensemble-mean variability collapses when Palmyra observations are unavailable.

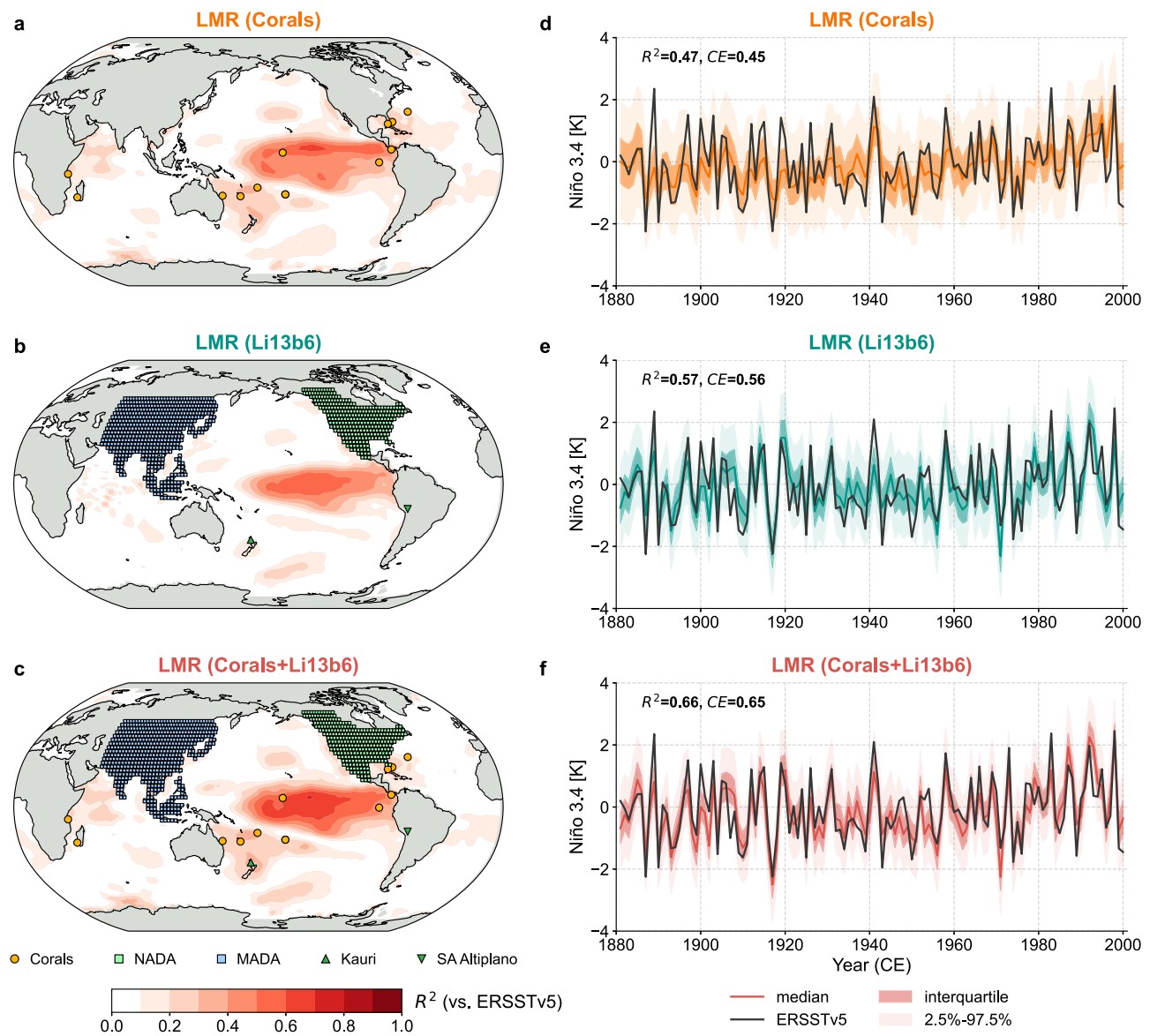

**Fig. 1 Validation of the last millennium reanalysis (LMR)[31,32] sea surface temperature reconstructions over the instrumental period (1881–2000 CE).**
The labels LMR (Corals), LMR (Li13b6), and LMR (Corals+Li13b6) denote the reconstructions assimilating corals reaching back before 1750 CE from the Ocean2k compilation[30], updated with the latest Palmyra data[28], the six best predictors from Li et al.[12] (denoted as Li13b6), and both data sources, respectively. **a–c** Spatial verification of the median field of the reconstructed boreal winter (December–February, DJF) surface temperature. Validation is performed against the Extended Reconstructed Sea Surface Temperature, Version 5 (ERSSTv5)[40]. The orange dots denote the location of the corals, the mint and blue squares denote the location of the North American Drought Atlas (NADA) (Version 2a)[41] and Monsoon Asia Drought Atlas (MADA)[42] sites, the green upward triangle denotes the location of the Kauri tree-ring composite[43], and the green downward triangle denotes the location of the South America Altiplano (SA Altiplano) tree-ring composite[44]. **d–f** Temporal verification of the median of the LMR reconstructed DJF Niño 3.4 series (colored curves) against the ERSSTv5 derived Niño 3.4 (black solid curve). For each reconstruction, dark shading denotes the interquartile range, and light shading denotes the central 95% region, from 2.5% to 97.5%. $R^2$ = coefficient of determination, $CE$ = coefficient of efficiency[70].

The assimilation of tree rings for Niño 3.4 reconstruction is challenging due to their distance to the target region. To overcome this, we gather the six best tree-ring based Niño 3.4 predictors previously identified by Li et al.[12] (denoted as Li13b6): the first two principal components of the North American Drought Atlas (NADA) (Version 2a)[41] and Monsoon Asia Drought Atlas (MADA)[42], the Kauri tree-ring composite[43], and the South America Altiplano tree-ring composite[44] (Supplementary Fig. 13), and assimilate them in LMR as proxies for Niño 3.4 SST (see "Methods"). The resultant reconstruction, denoted as LMR (Li13b6) in Fig. 1b, e displays skill with $R^2 = 0.57$ and $CE = 0.56$. The spatial verification (Fig. 1b) indicates skill peaking

around the center of the Niño 3.4 region and decaying quickly away from it. A major advantage of the tree-based reconstruction is its continuous nature over the past millennium (Fig. 2c).

We next assess the ENSO response to volcanism in both reconstructions. To perform an equitable comparison, we select large eruptions defined as volcanic stratospheric sulfur injection (VSSI) greater than 6 Tg S according to the eVolv2k version 3 dataset[45], with adjustments according to Toohey et al.[46] (see "Methods"), and focus on periods when the Palmyra coral record is available (Fig. 2a). We assess the forced ENSO responses using the widely-used superposed epoch analysis (SEA) approach (also known as compositing). The SEA (Fig. 3a, b) applied to the 12

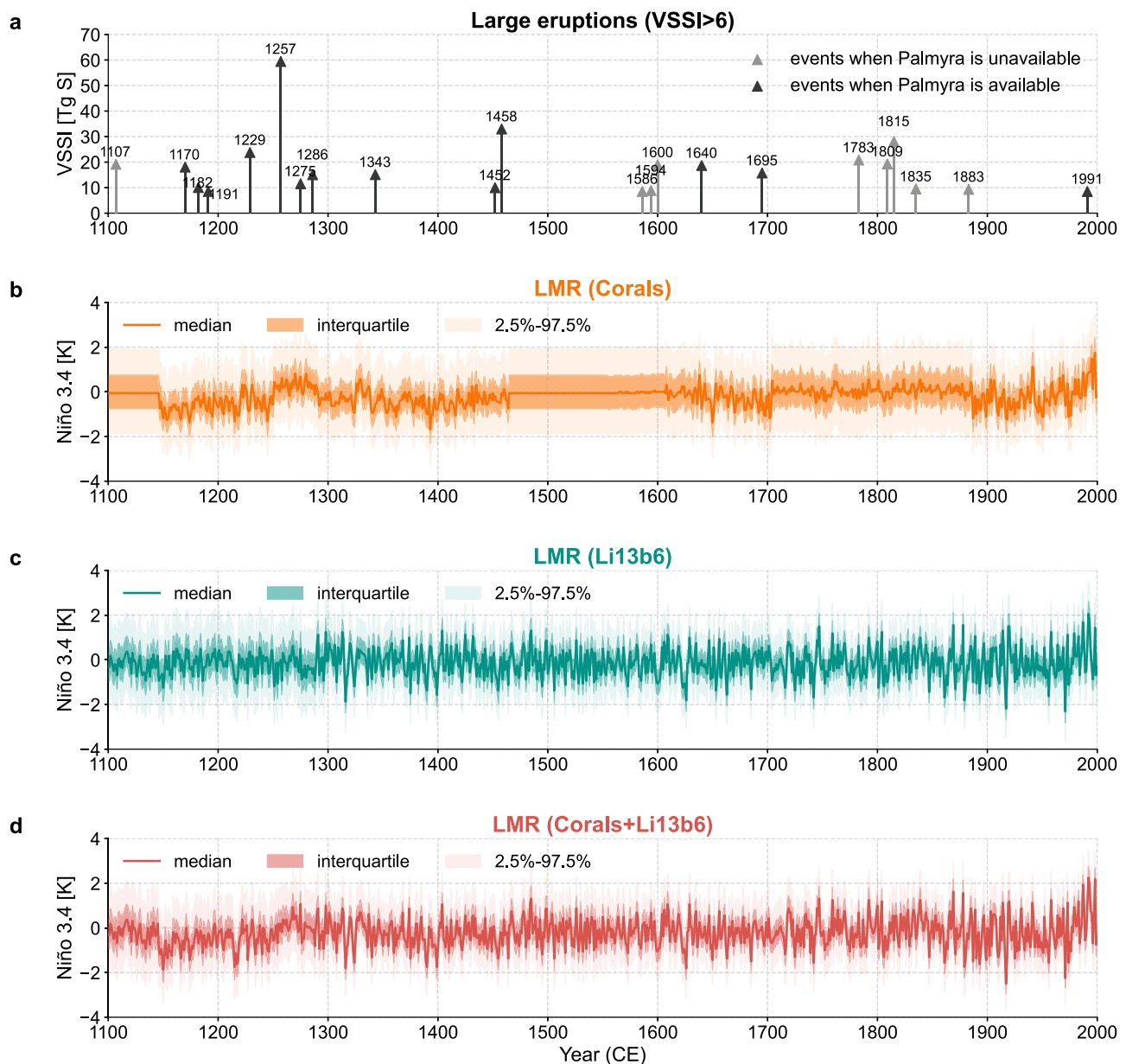

**Fig. 2 Timing of eruptions and ENSO events over 1100–2000 CE. a** The 22 largest eruption events, defined as a volcanic stratospheric sulfur injection (VSSI) greater than 6 according to eVolv2k version 3[45] and Toohey et al.[46]. The 13 events sampled by the Palmyra coral record[27,28] are colored in black, while other events are colored in gray. **b–d** Niño 3.4 reconstructions over the 1100–2000 CE period estimated with the Last Millennium Reanalysis (LMR) framework[31,32]. For each reconstruction, dark shading denotes the interquartile range, while light shading denotes the central 95% region (2.5% to 97.5% quantiles).

large events when Palmyra is available (Fig. 2a) suggests that there is no Niño 3.4 composite value that is significantly higher than that of randomly drawn non-volcanic years, even at the relatively permissive 90% level.

Therefore, SEA applied to both LMR (Corals) and LMR (Li13b6) suggests that the post-eruption ENSO response is not statistically different from non-eruption years.

**Combining corals and tree rings**. Given this agreement, we combine both coral and tree-ring archives in a single reconstruction, denoted as LMR (Corals+Li13b6). Verification statistics (Fig. 1c, f) show a slight skill improvement compared to both LMR (Corals) and LMR (Li13b6), with $R^2 = 0.66$ and $CE = 0.65$ against ERSSTv5. More importantly, the temporal gaps in LMR (Corals) over the past millennium are now filled with the

information from Li13b6 (Fig. 2), which allows a larger sample of eruption events, hence a more robust evaluation of ENSO's response to volcanism.

The SEA on LMR (Corals+Li13b6) suggests once again that there is no significant Niño 3.4 composite response in the year after eruptions (Fig. 3c, 4a), whether using the previously selected 13 events or all 22 large eruptions of the last millennium. In the latter case, however, the year 2 response breaches the 95% significance level (Fig. 4a), so we test its robustness in the next section.

**Impact of event selection**. While our results appear to reconcile previous results obtained from corals and tree rings, discrepancies remain in the published literature (e.g., Dee et al.[28], hereafter D20, vs Li et al.[12], hereafter Li13). Our best estimate, LMR

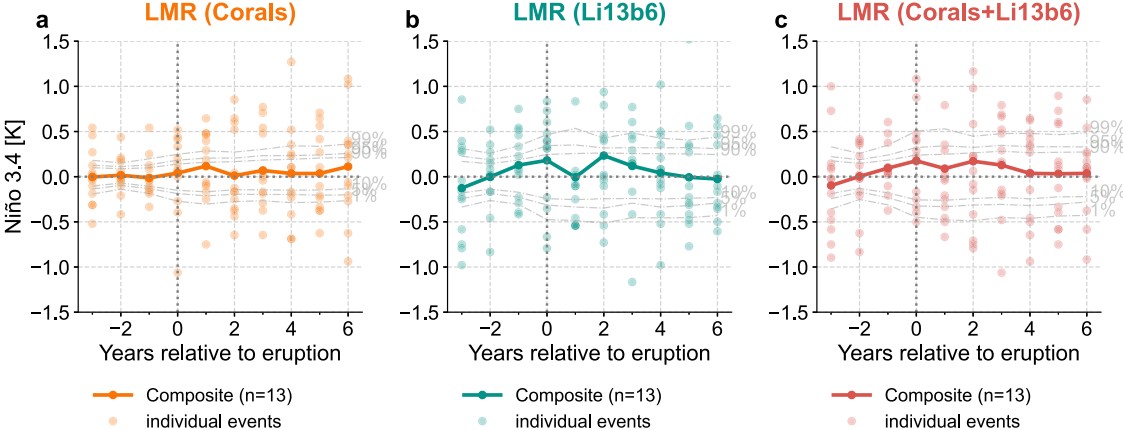

**Fig. 3 Superposed epoch analysis (SEA) of the last millennium reanalysis (LMR)[31,32] Niño 3.4 reconstructions around the 13 events when the Palmyra record is available. a** coral predictors only, **b** Li13b6 predictors only, and **c** corals+Li13b6 predictors. Solid curves with dark dots denote the composite mean, and the light dots denote the Niño 3.4 anomaly at each year for each individual event. The light gray dashed curves denote the 1%, 5%, 10%, 90%, 95%, and 99% quantiles of the composite means from 1000 bootstrap draws from non-volcanic years (see "Methods" for details).

**Fig. 4 A superposed epoch analysis (SEA) comparison between the last millennium reanalysis (LMR)[31,32] reconstruction and the Li et al.[12] reconstruction. a** SEA of the Niño 3.4 reconstruction LMR (Corals+Li13b6) regarding all the 22 events over the past millennium. **b** The large eruption events defined by different criteria: the volcanic explosivity index[47] greater than 4 (VEI > 4) and the volcanic stratospheric sulfur injection greater than 6 (VSSI > 6). **c–f** Superposed epoch analysis (SEA) of LMR (Corals+Li13b6) regarding the defined large eruption events based on VSSI > 6 over 1300–2000 CE, VSSI > 6 over 1300–1850 CE, VEI > 4 over 1300–2000 CE, and VEI > 4 over 1300–1850 CE, respectively. **g–j** Same as subplots (**c–f**), but for the Li et al.[12] reconstruction (denoted as Li13).

(Corals+Li13b6), is based on data sources used by both D20 and Li13, yet it shares the same conclusion as D20, at odds with Li13 as well as recent syntheses and simulations (Supplementary Text 7). What could account for such discrepancies?

We argue that a key and previously overlooked factor in such comparisons is the choice of events used to characterize the volcanic signal. Indeed, D20 selected events based on the stratospheric aerosol optical depth (SAOD) reconstruction in the eVolv2k dataset[45], which is essentially the same as our selection based on eVolv2k VSSI. Li13, on the other hand, defined events based on the volcanic explosivity index (VEI)[47]. Fig. 4b shows the differences in selected large eruptions by our criterion (VSS > 6) and the criterion used in Li13 (VEI > 4): (i) our selection includes more early events (pre-1300 CE) but fewer events from the instrumental era than Li13, whose selection starts in 1300 CE; (ii) even during the period of overlap, the two selections differ in detail, with only 1586, 1600, 1815, 1835, 1883, and 1991 CE selected by both sets of criteria.

To ensure a meaningful comparison, we apply SEA on both LMR (Corals+Li13b6) and the Li13 reconstructions using identical sets of events. Interestingly, applying the VSSI > 6 criterion over the 1300–2000 CE period, both reconstructions show insignificant year 1 ENSO responses (Fig. 4c, g), yet the year 2 response in LMR (Corals+Li13b6) breaches the 95% significance threshold. This response, however, vanishes when excluding eruptions from the instrumental era (Fig. 4d).

Using the VEI > 4 criterion over the 1300–2000 CE period favored by Li13, both reconstructions show significant year 1 responses, above the 99% quantile (Fig. 4e, i). Excluding events from the instrumental era, both reconstructions display more subdued responses in year 1, close to the 90% quantile (Fig. 4f, j). These results indicate that the two reconstructions (LMR Corals +Li13b6 and Li13) are essentially in agreement, and highlight event selection as the main source of divergence between the conclusions of D20, Li13, and recent syntheses (e.g.,[14]).

Which criterion is best suited to identifying the climate impacts of volcanism? Both VSSI and SAOD are derived from ice core sulfate deposition, a proxy for sulfate injection into the stratosphere, which directly affects the amount of shortwave radiation that enters the climate system[48]. VEI measures the volume of erupted tephra[47], which bears a less direct relation to climate forcing[48]. VSSI and SAOD would thus appear more justified than VEI for the study of the potential ENSO responses to volcanism. Comparing cases with and without the instrumental era events (Fig. 4c, e, i vs Fig. 4d, f, j) points to a drawback of SEA: like all composites, a small sample may be dominated by a small number of events with large anomalies, with most events showing a modest response[49]. In our case, the instrumental era events show an overall larger Niño 3.4 anomalies than pre-instrumental events, and come to dominate the SEA composites when they are included. This is true for both the LMR (Corals+Li13b6) and the Li13 reconstructions.

One possible reason for this outsize impact of recent eruptions is the non-stationary amplitude of the reconstructed indices, with a trend towards increasing amplitude towards the present (Supplementary Fig. 20). This trend could be real or an artifact of the reconstruction process, involving at least three factors. The first is data attrition, which reduces variability of a reconstruction going back in time since the proxy observations provide the only source of variability in our reconstructions. For instance, MADA is compiled using a correlation-weighted, ensemble-based modification of the point-by-point regression method[42], and the ensemble members would become less similar to each other during earlier periods, over which temporal variability is damped compared to recent intervals. The second factor is chronological uncertainty, a non-negligible problem for fossil corals[50] with errors compounding over time, potentially reducing the variability of composite series derived from them[36]. Finally, ENSO teleconnections may have changed over time, affecting the degree to which remote tree-ring sites capture ENSO variability[51]. An ENSO reconstruction based on the South American Altiplano composite[44] suggests that the non-stationary behavior of ENSO variance is not necessarily evident in tree-ring width chronologies. This is because when ENSO variance is low, the teleconnection is weak and tree ring variability is then more reflective of local temperature and moisture conditions than those from the remote tropical Pacific. Such non-stationarity may, however, additionally obscure the volcanoes-ENSO link.

Some of these factors can generate trends similar to those observed, as we show in Supplementary Text 6 (Supplementary Fig. 21d, e). However, the effects are not systematic and may not necessarily dampen the ENSO response to volcanism in our sensitivity experiments. Finally, in an artificial scenario where variance is stabilized over time, the year 1 ENSO response to pre-instrumental eruptions still emerges as unremarkable compared to non-volcanic years (Supplementary Fig. 22). We conclude that non-stationarity is not the leading reason behind the lack of a significant ENSO response prior to 1850 in our reconstructions.

In addition, we note that the instrumental period is quite short and devoid of the very large eruptions that mark the pre-instrumental Common Era, so results obtained over this period may not be statistically representative. Furthermore, instrumental records suggest a strong potential of coincidence between ENSO activity and volcanism, at least for the Agung (1963), El Chichon (1982), and Pinatubo (1991) eruptions, when strong El Niño events were already underway before the eruptions began[52]. Therefore, the choice of events against which to evaluate an ENSO-volcanism link appears to be the key variable determining whether a significant response is found and is strongly influenced by the small number of instrumental era eruptions.

## Discussion

Using state-of-the-art datasets and methods, combining the strengths of both coral and tree-ring records, the present evidence does not detect an effect of explosive volcanism on ENSO phase over the past millennium. This conclusion, consistent with D20[28], is now supported independently by both coral- and tree-ring-based reconstructions, and is robust to many methodological choices. We acknowledge that both coral and tree archives suffer from data attrition back in time, reducing reconstruction ensemble-mean signal variance back in time. Similarly, temporal gaps in the Palmyra coral record limit the pool of eruption events and non-volcanic years to a smaller size, making the comparison less stable. Beyond attrition, proxy archives harbor limitations of their own. The tree-ring records used here, for instance, are sensitive to the local hydrological expression of volcanic eruptions, which may be mistaken for the remote effects of ENSO via teleconnections[23]. The current coral network is more proximal to ENSO centers of action, yet is dominated by records from the western and central equatorial Pacific (Fig. 1a) which capture La Niña events more faithfully than El Niño events, and tend to underestimate the amplitude of large eastern Pacific (EP) events[36,38].

The sparse geographical coverage is another issue, preventing the estimation of tropics-wide SST in a way that would allow for a robust calculation of relative sea surface temperature (RSST)[18]. Indeed, it has been suggested that El Niño phases could be enhanced by volcanism even when the absolute SST signal in the central and eastern tropical Pacific is weak[53]. RSST highlights the impact of volcanism on ENSO relative to the tropical mean cooling, and may be computed from our reconstructed SST fields.

This is shown in Supplementary Fig. 12, which displays a temporally-flat tropical average temperature anomaly for most of the years between 1100–2000 CE, resulting in RSST-based Niño 3.4 anomalies that are indistinguishable from the SST-based ones. This is a direct result of the sparse coral network before the nineteenth century, and will not improve until this network is vastly expanded over tropical oceans.

Another caveat comes from limitations associated with the choice of the model prior in the LMR paleo-DA framework. The model prior here refers to the model simulation from which climate states are chosen at random by the ensemble Kalman filter. Its chief role is to provide the spatial covariance information within and between fields, and affects how the information from proxy observations affects locations where those observations are not available. A recent study[54] found that, in such a paleo-DA context, known model biases in the location of the South Pacific Convergence Zone (SPCZ) lead to incorrect inferences about Niño 3.4 SST from corals located in the SPCZ region. Moreover, they show that corals located in both the SPCZ and Niño 3.4 regions produce local cooling during the 1809 and 1815 eruptions, but all prior ensembles considered (including those drawn from 20th century reanalyses) have a covariance pattern that yields a remote influence (that is, the influence of one region on another) inconsistent with this pattern. In addition to model bias, this suggests that model priors conditional on eruption time would be important for better capturing the effects of volcanism encoded in proxy records. We note that, in the present study, these biases are mitigated prior to 1800, when only proxies tied to the Niño 3.4 index are used, and spatial covariances from the model prior have a minimal impact.

Finally, a recent modeling study[25] suggests that the ENSO response to volcanism is rather weak during DJF because it changes sign around that season. Yet, the response could be strong before the sign changes, which is usually during July–September (JAS) and/or October–December (OND). Repeating our analysis using JAS and OND as target seasons, we find our conclusions insensitive to this choice (Supplementary Text 4). This is in apparent contradiction to the results of ref. [25], yet may be explained by the limitations of currently available proxy records noted above. Assigning eruption events to specific years based on volcanic forcing reconstructions could also lead to time offsets in analyses using different target seasons[55], which will potentially obscure the relation to eruption events. This problem can only be resolved with a better knowledge of eruption season than presently available.

Multiple studies have highlighted the importance of the spatial distribution (particularly, the inter-hemispheric asymmetry) of stratospheric aerosol loadings in the sign of the ENSO response[21–23,56]. This distribution is also difficult to constrain precisely from the sparse network of ice cores used in forcing reconstructions like eVolv2k. In sum, the absence of a consistent ENSO response in the relatively small statistical sample of the last millennium does not necessarily imply a lack of ENSO predictability associated with explosive volcanism. As suggested by recent modeling studies[17,25], the forcing magnitude, location, and season of the eruption, as well as pre-conditioning of the ENSO state can greatly affect the ENSO response to volcanic eruptions. These multiple factors must align to favor the development of ENSO events, providing a source of predictability only when these factors are known with a sufficient degree of accuracy. It is therefore reasonable to expect that the lack of an observed, consistent relationship between ENSO phases and explosive volcanism in our reconstructions may be due, in part, to an imperfect knowledge of these factors. However, even when controlling for eruption timing in PMIP3 and CESM-LME simulations, the SEA still demonstrates inconsistent ENSO responses to volcanism when this major source of variability is held fixed (Supplementary Text 3), suggesting that all factors need to be jointly determined, or that a larger ensemble is needed to discern common trends.

Given the large number of degrees of freedom, a correspondingly large sample size is needed to isolate a consistent signal – larger perhaps than offered by the past millennium. It is thus important to develop more high-resolution proxy records spanning the tropical oceans over the last millennium, and possibly extend them through the longer Holocene. The reconstruction of volcanic forcing also needs to be expanded with longer temporal coverage for more robust statistics[57]. Until these goals have been achieved, it is unclear how much one can confidently conclude from the current paleoclimate record about the contribution of volcanic eruptions to ENSO dynamics, or to the assessment of the risks posed by solar radiation management strategies in relation to ENSO.

## Methods

**Paleoclimate data assimilation.** The version of the Last Millennium Reanalysis (LMR) paleoclimate data assimilation (paleo-DA) framework used here[31,32] is an offline ensemble Kalman filter[58], optimized for multivariate climate field reconstruction[59]. It consists of a collection of prior states generated by the CCSM4 climate model[60], a proxy database, a set of forward operators or proxy system models (PSMs)[61] that translate the environmental variables to proxy units, and an ensemble Kalman filter operator. In our implementation, the temporal variation of the posterior stems entirely from the temporal information from the proxies, while the covariance structure of the model prior serves to spread the temporal information of each proxy to remote regions and other variables than those directly related to the proxy. Proxy error variance is estimated as the mean squared error of the PSM output over the calibration period. For further details, see previous studies[31,32,59]. For computational convenience, this study utilizes a fast implementation of the LMR framework, LMRt[62]. In each assimilation experiment, we perform 50 Monte Carlo iterations, each using a different randomly chosen 100-member ensemble states from the CCSM4 last millennium simulation[60] as the model prior. No proxy randomization is performed to guarantee the similarity between each ensemble member so that the median curve of the reconstructed Niño 3.4 index series is representative of the whole reconstruction product. The default covariance localization[63] radius of 25,000 km is applied, and we note that the results are broadly insensitive to this choice.

Note that the calibration period for the PSMs is 1850–2000 CE so as to achieve the best reconstruction skill. To guard against potential overfitting, as well as the potential impact of climate change seen in NADA PC2 and the Kauri composite (Supplementary Fig. 13d, e), a cross-validation of the reconstructed Niño 3.4 is performed with disjoint calibration and validation periods (see Supplementary Text 1); the results show that reconstruction skill is stable to this choice.

**Data sources.** We consider information from seasonally-sensitive or monthly-resolved proxy records. Coral records are from the Ocean2k compilation[30] updated with the latest Palmyra data[28]. Each coral record is treated as a proxy for local sea surface temperature (SST), and is calibrated over 1850–2000 CE through a univariate linear regression procedure against the local, boreal winter (DJF) SST, which shows high reconstruction skill as in previous studies[31,32]. Experiments with a model that takes the oxygen isotopic composition of seawater into account did not produce noticeable improvements, which is in agreement with a recent study[54]. Tree-based records from both hemispheres are taken from a previous reconstruction (Li13[12]), using seven predictor timeseries. The six best predictors are the first two principal components (PCs) of North American Drought Atlas (NADA) (Version 2a)[41] and Monsoon Asia Drought Atlas (MADA)[42], the Kauri tree-ring composite[43], as well as the South American Altiplano composite[44], with the explained variance of Li13 being 11.2%, 8.4%, 40.5%, 24.2%, 38.1%, 56.5%, respectively. The other predictor is the west-central Argentina composite[64], which contributes negligibly to reconstruction skill (with an explained variance of Li13 of 1.8%), and is ignored here. We refer to these best six records as Li13b6.

We reproduced the principal components of NADA and MADA via principal component analysis (PCA), so that we are able to extend the timespan of NADA PCs to 1001–2000 CE, compared to the original timespan (1300–2000 CE) in Li13. The timespan of the reproduced MADA PCs remains 1300–2000 CE due to the temporal coverage of MADA itself[42]. The Kauri tree-ring composite covers the 1578–2003 period, and the South American Altiplano tree-ring composite covers 1290–2010 CE. Data beyond 2000 CE are not used since our reconstruction stops at 2000 CE. Note that the South American Altiplano composite we use is the residual chronology instead of the standard chronology. A residual chronology is obtained by filtering (prewhitening) the standard chronology via autoregressive modeling[65]. This procedure removes some low-frequency variability and therefore emphasizes the higher-frequency variability in tree-ring growth and remains appropriate for the reconstruction of interannual variability. All six predictors are treated as

proxies for Niño3.4 SST, following the idea of Li et al.[12], hence effectively no localization is applied in practice. They are calibrated over 1850–2000 CE through a univariate linear regression procedure against the Niño3.4 index series derived from a spatially completed version of HadCRUT4.6[66] leveraging the GraphEM[67] algorithm. Supplementary Fig. 13g–i show a temporal verification of each predictor against the Niño 3.4 series derived from ERSSTv5 as in Fig. 1d–f.

**Superposed epoch analysis**. Superposed epoch analysis (SEA)[68] is a widely-used method for analyzing the climate responses to volcanic eruptions[12,28,69]. In our case, we extract the median Niño3.4 index series as the target for analysis for each LMR reconstruction. Large eruption event years (Year 0) are defined as years with volcanic stratospheric sulfur injection (VSSI) greater than 6 Tg S according to the eVolv2k version 3 dataset[45], with adjustments according to Toohey et al.[46], in which dating uncertainties (+/− 2 years) of eruption years are tested and fixed based on maximizing the 3 year extratropical post-eruption cooling. We choose the threshold 6 Tg S so that the selected large eruptions are consistent with those of D20[28], who chose a threshold based on stratospheric aerosol optical depth (SAOD > 0.07). Note that since the reconstructions are annualized over the December–February (DJF) season, Year 0 will be before the eruptions if they have occurred after February, and we therefore should mainly look at Year 1 when analyzing the post-eruption responses.

SEA considers segments around each event, here extending from 3 years prior to the event year and 6 years after the event year. As in previous work[28], the mean of the 3 years prior to the event year is removed so that each segment represents the anomaly response relative to the mean state before each event. A composite is obtained by averaging these 10-year windows. A bootstrap sampling distribution is then obtained: the same number of years as the eruption events under consideration are randomly drawn from the pool of non-volcanic years 1000 times, and the composite is calculated for each draw with an identical process used to calculate the composite for eruption years, based on which we calculate the 1%, 5%, 10%, 90%, 95%, and 99% quantiles of the non-volcanic years as the significance levels.

SEA is sensitive to dating uncertainties, as the compositing procedure requires that the reconstruction segments of individual events align precisely, so minor timing offsets can compound or damp the composite signal[50]. SEA may also be affected by the small sample statistics. This may first come into play through a type I error, which is tested in the Supplementary Text 2. We find that even small ensembles (n = 5) can mitigate this problem, so it is not material for this particular issue.

## Data availability

The LMR reconstructions generated in this study are available at https://doi.org/10.5281/zenodo.5716165. All the required information is at https://zenodo.org/record/5893781.

## Code availability

The code that supports the findings of this study is available at https://doi.org/10.5281/zenodo.5716165.

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

## Acknowledgements

The authors acknowledge support from the Climate Program Office of the National Oceanographic and Atmospheric Administration (grants NA18OAR4310426 to J.E.G., NA18OAR4310422 to G.J.H., and NA18OAR4310420 to K.J.A.). GJH also acknowledges support from the NSF through grant AGS–1702423. M.S.M. was supported by the National Agency for the Promotion of Science and Technology (PICT 2013-1880 and PICT 2018-03691) and the National Scientific and Technical Research Council of Argentina (PIP CONICET 11220130100584).

## Author contributions

J.E.G., K.J.A., G.J.H., A.T.W., and F.Z. designed the research. F.Z. conducted all calculations and produced all the figures. All authors (F.Z., J.E.G., K.J.A., G.J.H., A.T.W., M.S.M., M.T., J.K.) interpreted results and wrote the paper.

## Competing interests

The authors declare no competing interests.
