## [Peer Review File · Nature Communications]

REVIEWER COMMENTS

Reviewer #1 (Remarks to the Author):

The manuscript investigates the relationship between volcanism and ENSO using a framework of LMR paleoclimate data assimilation. This study finds that there is no clear statistical relationship between them. This result is in accord with Dee et al. (2020) but contradicts with Li et al. (2013). The study also shows that the significance of ENSO response to volcanism is sensitive to the choice of eruptions, suggesting the nonstationary response of ENSO to volcanism.

The manuscript is generally well written, well organized, and the analyses are thorough. The topic would be interesting to many climate scientists and appropriate for Nature communications.

That said, however, I feel that the manuscript lack novelty and some of the conclusions are not adequately supported by data. One of their main conclusions showing the weak relationship between ENSO and volcanism has been already shown by Dee et al. (2020), which is reasonable as this study largely relies on the data used by Dee et al. (2020) (i.e. coral data from Palmyra atoll). The other finding (i.e. non-stationary response of ENSO to volcanism) would be interesting and novel, but is not well supported by data: this may be due to the data attrition, as the authors mentioned. In my opinion, more samples should be necessary to conclude. This is true for the conclusion about the asymmetry of volcanic forcing and ENSO.

Considering that it would require massive efforts to increase the number of samples, I suggest that the authors should submit the manuscript to another journal. I hope the comment below helps improve the manuscript.

L86-88: How did you localize information when you assimilate the PCs? In case no localization is applied to the PCs, are there good reasons to do so?

L90-94: Is this the right interpretation? The temporal variation in LMR is fully from observations assimilated. If all the observations tend to underestimate the NINO3.4 anomaly, the assimilated product should also underestimate it.

L140: How sensitive is the result to the choice of the year after which eruptions are excluded? Isn't it subjective to choose the year 1850CE?

L148-149: The authors may want to do a pseudo proxy experiment to investigate if the result is because of the data attrition.

Reviewer #2 (Remarks to the Author):

Review of "Volcanoes and ENSO: a re-appraisal with the Last Millennium Reanalysis" by Zhu et al

Summary and recommendation

Volcanic forcing is one of the most fundamental natural forcings on climate, which modulates internal variability at interannual to centennial time scales. This work integrates the paleoclimate data into a consistent dynamical framework and re-appraises the volcano-El Niño relationship by comparing a set of tree-ring and coral based observations. I acknowledge that method of this study is novel and may provide new insights on the volcano-El Niño relationship. However, current analysis and results are dubious in a couple of ways as shown below. Therefore, a MAJOR revision is needed before this work can be published in Nature Communications.

My major concern involves the use of global tree rings to reconstruct the ENSO signal. Even during the last century when instrumental observations are available, the relationship between the Indian summer monsoon and ENSO varies greatly over time, especially a dramatic reduction after the

1970s (Kumar et al. 1999 Science). Similar non-stationarity of the relationship with ENSO exists for the East Asian summer monsoon as well (Wang et al. 2008 JC). Based on the stable Pacific-North American teleconnection, previous studies reconstructed the ENSO index from North American tree rings (Li et al. 2011 NCC; Cook et al. 2008; D'Arrigo et al. 2005 GRL). Will there be any significant El Niño responses to the tropical eruptions when NA tree-ring based ENSO reconstructions were used? Further analysis on this may help clarify the paleo volcano-El Niño relationship.

Another concern is about the results reported in Fig. 1. What does year 0 mean here? In observations El Niño occurred in December of 1902, 1913, 1963, 1982, 1991, right after the eruptions. However, negative Niño 3.4 indices are observed in December of these years (Figs. 1g, 1h, and 1i), which is inconsistent with the observations.

Line.37: The Kelvin wave response to suppressed West African monsoon belongs to tropical teleconnection.

Line. 117: In your work year 0 actually refers to the ENSO before the eruptions.

Line.141-142: Why does the inclusion of instrumental-era events lead to biased results? There are at least 6 warm responses among 7 eruptions (Figs. 1g, h and i). The weak relationship may be induced by the unstable teleconnection of ENSO in monsoon Asia. If so, the conclusion in Line. 148 is unjustified. Further analysis is needed in this regard.

Responses to Reviewers

We are grateful to both referees for their perceptive comments, which help us refine our arguments and improve the manuscript's quality. In the following, we address each comment in turn.

To Reviewer I

1 Comments

1.1

The manuscript is generally well written, well organized, and the analyses are thorough. The topic would be interesting to many climate scientists and appropriate for Nature communications. That said, however, I feel that the manuscript lack novelty and some of the conclusions are not adequately supported by data. One of their main conclusions showing the weak relationship between ENSO and volcanism has been already shown by Dee et al. (2020), which is reasonable as this study largely relies on the data used by Dee et al. (2020) (i.e. coral data from Palmyra atoll). The other finding (i.e. non-stationary response of ENSO to volcanism) would be interesting and novel, but is not well supported by data: this may be due to the data attrition, as the authors mentioned. In my opinion, more samples should be necessary to conclude. This is true for the conclusion about the asymmetry of volcanic forcing and ENSO.

Response: We thank the reviewer for the comments. With respect to the novelty of our study:

- While our does utilize the same Palmyra record as Dee et al. (2020), it also leverages the tree-ring based ENSO predictors used in Li et al. (2013). This is, to the best of our knowledge, the first time these predictors are married and their influences compared systematically, resulting in remarkably high validation statistics compared to other existing Niño 3.4 reconstructions (Extended Data Fig. 1 in the main text);
- In the revised manuscript, we found that the major reason that our reconstruction [LMR (Corals+Li13b6) in the revised manuscript] and the Li et al. (2013) reconstruction (hereafter Li13) lead to seemingly divergent conclusions is not the composition of data sources, but the selection of eruption events for analysis: if the selection based on volcanic sulfur injection ($VSSI > 6$) is applied, both our reconstruction and the Li13 reconstruction show similar insignificant post-eruption ENSO responses; if the selection based on explosivity ($VEI > 4$) is applied, both reconstruction show similar significant post-eruption ENSO responses (this finding is presented in the revised Fig. 4 in the main text). Therefore, our reconstruction and Li13 are actually in agreement with each other, a significant result unto itself. We note that the selection of large eruptions is the first order factor to the conclusion on the volcanoes-ENSO linkage that has been overlooked in published literature, including both D20 and Li13. We also discussed why VSSI might be a more justified criterion than VEI.
- We agree that more samples would be essential for stronger conclusions, and have softened the tone. The comparison of SEA on eruption sets with and without instrumental events (presented in the revised Fig. 4 in the main text) reveals how sensitive the SEA method could be when the sample size is small, which calls for caution when a robust relationship is claimed based on such analysis.

In regards to nonstationarity, we agree that data attrition (as the Reviewer suggests) is one potential issue, yet we note that it cannot be the primary factor, as the Li13 reconstruction features constant availability over this period, yet it still shows nonstationarity and a seemingly divergent conclusion is obtained. Nevertheless, our revised manuscript thoroughly explores the issue and its potential impact on our conclusions:

- To mitigate against the nonstationarity in the reconstructed Niño 3.4 series, we have updated all our reconstructions, assimilating corals that reach back before 1750 CE, instead of the whole Ocean2k collection, which included many short corals that. With these long coral records, the skill of the reconstructions is more homogenous over time, so that the skill over the instrumental era may represent the skill of the last millennium more faithfully, although we note that the issue still exists to some extent, as it does in all existing reconstructions.
- We conducted a series of pseudoproxy experiments (PPEs) to investigate the origins of the observed nonstationarity and the impact on our SEA, and we put the details in the response to Comment 1.5.

We absolutely agree with the reviewer that more samples are needed to get reliable conclusions, and we have conveyed this idea in the end of our main text that “Until these goals have been achieved, it is unclear how much one can conclude from the paleoclimate record about the contribution of volcanic eruptions to ENSO dynamics, or to the assessment of the risks posed by solar radiation management strategies in relation to ENSO.” In addition, based on the same reasoning, every existing study should be cautious when observational studies claim robust volcanoes-ENSO relationship when there are a declining number of proxies available back through time.

Please note that the ranking analysis in our first submission is now removed, because it looked at absolute Niño 3.4 anomalies after each eruption, which – we concluded – is not necessarily the best measure of the impact of volcanism. The impact of volcanism would ideally be measured as the difference between the state with volcanic forcing and the state without the forcing. Since this counterfactual is not available to us from the real world, we have to measure the difference by comparing the distribution of the responses in volcanic years and that of the responses in non-volcanic years. In this sense, SEA makes the most sense, despite its other caveats, so we continue to use SEA. For the same reason, the analysis of asymmetries is also removed.

1.2

L86-88: How did you localize information when you assimilate the PCs? In case no localization is applied to the PCs, are there good reasons to do so?

Response: Thank you for pointing this out. The PCs, along with other tree-ring based predictors, have been used as direct ENSO predictors in Li et al. (2013) and others before them, and we followed this idea to treat them as direct ENSO proxies as well, instead of the proxy of local environments in our data assimilation experiments. Therefore, in practice their potential contribution is not a function of their spatial distance from the Niño 3.4 box, so effectively there is no localization for these predictors. We have made this clearer in the “Data Sources” section (L389-390) in the main text.

1.3

L90-94: Is this the right interpretation? The temporal variation in LMR is fully from observations assimilated. If all the observations tend to underestimate the NINO3.4 anomaly, the assimilated product should also underestimate it.

Response: Thank you for raising this good question. We confirm that the interpretation is correct. Data assimilation is a process of optimal estimation of the states. Even if each observation has relatively large error, processing them through the Kalman filter can yield overall better analysis than any single observation, although we agree that underestimation may still persist to some extent.

Also, please note that since we do not use the VEI criterion used in Li13, we have removed the scatter plots in the revised Fig. 1, Extended Data Fig. 2, and SI Figs. 11-14, where originally we plotted the colored dots to represent the instrumental era eruptions used in Li13.

1.4

L140: How sensitive is the result to the choice of the year after which eruptions are excluded? Isn't it subjective to choose the year 1850CE?

Response: The reason that we drew a boundary at 1850 CE was not subjective. We found the instrumental era eruptions show overall stronger responses to volcanism compared to the pre-instrumental era events, hence dominating the SEA result. Yet our conclusion is not anchored to this comparison alone. As mentioned in the response to Comment 1.1, we compared the criteria based on VSSI and VEI, and found that our reconstruction LMR (Corals+Li13b6) and the Li13 reconstruction agree with each other as long as the consistent event set is used in the SEA. Overall, we found that the selection of events overall is a first order factor that affects the conclusion. This is presented in the revised Fig. 4 and the revised text L114-168 of the main text.

1.5

L148-149: The authors may want to do a pseudo proxy experiment to investigate if the result is because of the data attrition.

Response: Thank you for the suggestion. We have now conducted a set of pseudoproxy experiments (PPEs) to test if the nonstationarity we observe in our reconstructions is due to data attrition, and the results are presented in the revised SI Text 6.

The results indicate that the data attrition caused by temporal availability is unlikely the cause of the observed nonstationarity in the ENSO-volcanism relationship (see the revised SI Fig. 17b,g against a,f). In addition, adding proxy noise uniformly to the proxy database with signal-to-noise ratio (SNR) equals to 1 may help reduce the significance level of the year 1 responses by around 5% (SI Fig. 17c,h against b,g). Dating uncertainties in coral proxies could help create the observed nonstationarity (SI Fig. 17d,i against c,h), but it may or may not be able to hold a high significance level for the year 1 responses in the meantime (SI Fig. 17d,i show significant year 1 responses with the nonstationary reconstructed Niño 3.4 series, while SI Fig. 17e,j show insignificant year 1 responses with the nonstationary reconstructed Niño 3.4 series), suggesting that the observed nonstationarity can be a combination effect of multiple factors such as proxy error and proxy dating uncertainty, yet the impact of such nonstationarity on the year 1 responses is not systematic.

To Reviewer II

2 Comments

2.1

My major concern involves the use of global tree rings to reconstruct the ENSO signal. Even during the last century when instrumental observations are available, the relationship between the Indian summer monsoon and ENSO varies greatly over time, especially a dramatic reduction after the 1970s (Kumar et al. 1999 Science). Similar non-stationarity of the relationship with ENSO exists for the East Asian summer monsoon as well (Wang et al. 2008 JC). Based on the stable Pacific-North American teleconnection, previous studies reconstructed the ENSO index from North American tree rings (Li et al. 2011 NCC; Cook et al. 2008; D'Arrigo et al. 2005 GRL). Will there be any significant El Niño responses to the tropical eruptions when NA tree-ring based ENSO reconstructions were used? Further analysis on this may help clarify the paleo volcano-El Niño relationship.

Response: We agree with the reviewer that the ENSO teleconnections that tree-ring records rely on when used for ENSO reconstruction could be unstable, and we have mentioned this caveat in L185-188 of our original manuscript. Indeed, this concern is true for all tree-ring based reconstructions of ENSO variability, including Li et al. (2013) amongst other used to suggest a link between ENSO and volcanism. Our study aims to investigate the reason why coral-based studies (e.g. D20) and tree-ring records based studies (e.g. Li13) lead to seemingly divergent conclusions, so we followed Li13 and assimilated the same predictors used in Li13, which also include the North America trees as the reviewer suggested, but in the form of NADA PCs.

With the caveat of unstable ENSO teleconnections, Li13 claimed a robust volcanoes-ENSO relationship. However, we found that such significant post-eruption responses are tied to their selection of large eruption events based on explosivity ($VEI > 4$), which measures the volume of the erupted tephra (Newhall and Self, 1982) that is not directly linked to the climate system response. With the same selection of the events, we get similar significant year 1 response with our own reconstruction (see revised Fig. 4 in the main text). However, we would argue that the stratospheric sulfur injection that VSSI measures might be a more justified metric since the amount of the stratospheric sulfur injection affects the solar radiation that enters the climate system directly. With the criterion of $VSSI > 6$, we see similar insignificant year 1 response in both our reconstruction and the Li13 reconstruction (see revised Fig. 4 in the main text).

Even if we accept the $VEI > 4$ criterion, we found that the significance originates primarily from the instrumental era events, among which at least three of them could occur simply by coincidence, as we mentioned in L157-159 of the original manuscript, "instrumental records have shown evidence of the potential of coincidence between ENSO activity and volcanism, at least for the Agung (1963), El Chichon (1982), and Pinatubo (1991) eruptions, when strong El Niño events were already underway before the eruptions went off."

We agree that for the various reasons discussed in the text, we cannot completely rule out that a volcanoes-ENSO relationship might exist. Our study however strongly cautions against the claimed robust relationship in previous studies given all the factors discussed in the manuscript and various caveats.

2.2

Another concern is about the results reported in Fig. 1. What does year 0 mean here? In observations El Niño occurred in December of 1902, 1913, 1963, 1982, 1991, right after the eruptions. However, negative Niño 3.4 indices are observed in December of these years (Figs. 1g, 1h, and 1i), which is inconsistent with the observations.

Response: Thank you for pointing this out, and sorry for the confusion. We took that convention and labeled the time axis of our DJF reconstruction based on the year of January, following Li et al. (2013) and other existing literature (e.g. Wilson et al., 2010; Emile-Geay et al., 2013). That is, year 0 of 1902 refers to the January of 1902 along with its previous December in 1901 and its following February in 1902. We have revised the text in L77-78 of the main text to make this clearer. Therefore, the reviewer is correct (in Comment 2.4) that year 0 refers to the ENSO before the eruptions if the eruptions occur after February. That is why we should mainly look at the year 1 response, and we did so in our revision.

One may, of course, label the DJF reconstruction based on the year of December, and that will lead to a -1 shift to both our reconstructions and the Li13 reconstruction, making the year 0 response our target instead. In addition to this not being a common convention for winter seasons that span the calendar year, a potential disadvantage of that choice is that for eruptions occur early in the year, say March, the year 0 response will be already 9 months later after the eruption, and that response could already be weakened if there ever exists strong response.

Also, please note that since we do not use the VEI criterion used in Li13, we have removed the scatter plots in the revised Fig. 1, Extended Data Fig. 2, and SI Figs. 11-14, where originally we plotted the colored dots to represent the instrumental era eruptions used in Li13.

2.3

L37: The Kelvin wave response to suppressed West African monsoon belongs to tropical teleconnection.

Response: Thank you for pointing this out. It has been fixed in the revised text in L38 of the main text.

2.4

L117: In your work year 0 actually refers to the ENSO before the eruptions.

Response: The reviewer is correct, and as discussed in the response to Comment 2.2, we have revised the text in L78-79 of the main text to make this clearer.

2.5

L141-142: Why does the inclusion of instrumental-era events lead to biased results? There are at least 6 warm responses among 7 eruptions (Figs. 1g, h and i). The weak relationship may be induced by the unstable teleconnection of ENSO in monsoon Asia. If so, the conclusion in Line. 148 is unjustified. Further analysis is needed in this regard.

Response: As we mentioned in L157-159 of the original manuscript, “instrumental records have shown evidence of the potential of coincidence between ENSO activity and volcanism, at least for the Agung (1963), El Chichon (1982), and Pinatubo (1991) eruptions, when strong El Niño events were already underway before the eruptions went off.” Moreover, these are statistics based on a small sized sample during the instrumental era, we thus cannot claim that the relationship is robust.

With respect to the potential nonstationarity of the ENSO teleconnection, failure of teleconnection during a period of weak ENSO activity may be more strongly expressed in local/regional tree-ring reconstructions, but this issue can be solved by using a large network of ENSO proxies such as those used in this study: tree-ring width chronologies distributed over major ENSO teleconnection regions, including the North America region as

the reviewer suggested, and a large network of coral data in the tropical Pacific. The usage of such a large multi-proxy network makes the ENSO reconstruction more reliable than those purely based on tree-ring networks, and minimizes the possible influence of local climate variability not related to ENSO on tree growth during periods of low ENSO activity.

References

- Emile-Geay, J., K. M. Cobb, M. E. Mann, and A. T. Wittenberg, 2013: Estimating Central Equatorial Pacific SST Variability over the Past Millennium. Part II: Reconstructions and Implications. *Journal of Climate*, **26** (7), 2329–2352, doi:10.1175/JCLI-D-11-00511.1, URL <https://journals.ametsoc.org/jcli/article/26/7/2329/33095/Estimating-Central-Equatorial-Pacific-SST>.
- Li, J., and Coauthors, 2013: El Niño modulations over the past seven centuries. *Nature Climate Change*, **3** (9), 822–826, doi:10.1038/nclimate1936, URL <https://www.nature.com/articles/nclimate1936>.
- Newhall, C. G., and S. Self, 1982: The volcanic explosivity index (VEI) an estimate of explosive magnitude for historical volcanism. *Journal of Geophysical Research: Oceans*, **87** (C2), 1231–1238, doi:10.1029/JC087iC02p01231, URL <https://agupubs.onlinelibrary.wiley.com/doi/abs/10.1029/JC087iC02p01231>.
- Wilson, R., E. Cook, R. D'Arrigo, N. Riedwyl, M. N. Evans, A. Tudhope, and R. Allan, 2010: Reconstructing ENSO: the influence of method, proxy data, climate forcing and teleconnections. *Journal of Quaternary Science*, **25** (1), 62–78, doi:10.1002/jqs.1297, URL <http://dx.doi.org/10.1002/jqs.1297>.

REVIEWERS' COMMENTS

Reviewer #1 (Remarks to the Author):

I am sorry for my late review. The questions I raised are sufficiently mentioned in the revised manuscript. I have no further questions or comments.

Reviewer #2 (Remarks to the Author):

I appreciated the authors' effort to address all of my concerns. Although I don't agree the argument "instrumental records have shown evidence of the potential of coincidence between ENSO activity and volcanism, at least for the Agung (1963), El Chichon (1982), and Pinatubo (1991) eruptions, when strong El Niño events were already underway before the eruptions went off", since large eruption can certainly modulate the development of an El Niño through ocean dynamic thermostat, land-sea thermal contrast, and equatorward migration of the ITCZ (Robock 2020Science), I think this paper can stimulate the discussions on this open question and comparison between proxy analyses and model simulations, thus I recommend an Accept of this paper in Nature Communications.